# Carnot and the Archetype of Waterfalls

**DOI:** 10.3390/e26121066

**Published:** 2024-12-07

**Authors:** Hans U. Fuchs, Elisabeth Dumont, Federico Corni

**Affiliations:** 1Center for Narrative in Science, 8400 Winterthur, Switzerland; hans.fuchs@narrativescience.org; 2Faculty of Education, Free University of Bolzano, 39042 Bressanone, Italy; 3School of Engineering, Zurich University of Applied Sciences at Winterthur, 8401 Winterthur, Switzerland; dumo@zhaw.ch

**Keywords:** Carnot, Waterfall Analogy, heat engines, thermal tension, caloric, power, generalized energy principle, metaphor, analogy

## Abstract

Carnot treats Heat as a Force of Nature, with its typical fundamental characteristics of intensity and thermal tension (temperature and temperature difference), extension (amount of heat, i.e., caloric), and power. To suggest how the three aspects are related, he applies the imagery of waterfalls to causative thermal processes: heat powers motion in a heat engine just as falling water does when activating rotation in a water wheel. We understand Carnot’s waterfall imagery as an archetype of human reasoning—as an embodiment of how we experience and understand causative (agentive) phenomena. We project it onto the macroscopic phenomena identified in physical science and so unlock the power of analogical structure mapping between theories of fluids, electricity and magnetism, heat, substances, gravity, and linear and rotational motion. In particular, the notion of (motive) power of a waterfall lets us create imaginative explanations of the interactions of Forces of Nature and helps us construct a generalized energy principle. Two-hundred years after Carnot made us aware of it, his Waterfall Analogy is a powerful example of theory construction with roots deep in how we experience phenomena as caused by natural agents.

## 1. Introduction and Overview

To Carnot, Heat presented itself as a Force of Nature ([1], pp. 1–2), with properties that could be apprehended by comparing it to the archetypal causative phenomenon of a waterfall ([1], p. 28). This was the source of his suggestion that Heat powers Motion in a heat engine just as falling water does in the case of a water wheel.

### 1.1. The Waterfall Analogy

Carnot compared the height of the fall of water to the temperature difference between furnace and cooler, and the quantity of water falling from the high point to the lower one to the amount of heat falling from the hot furnace to the cold cooler, from where it flows into the environment. His *Waterfall Analogy* suggested to him the idea that the power of the Force of Heat could be proportional to the product of the two measures—in analogy to how the power of a waterfall is proportional to the product of height of fall and flow of water. Here are his words ([1], p. 28; the translations here and in the rest of the paper are ours):


*“According to the notions established up to now, we can compare with some accuracy the motive power of heat to that of a waterfall: both have a maximum that cannot be exceeded, whatever the machine used to receive the action of the water, and whatever the substance used to receive the action of the heat. The motive power of a waterfall depends on its height and the quantity of liquid; the motive power of the heat also depends on the quantity of caloric employed and on what could be called, in effect, the height of its fall [here comes Carnot’s Footnote 1, see Section 3.2], that is to say, on the difference in temperature of the bodies between which the transfer of caloric takes place. In the waterfall, the motive power is strictly proportional to the difference in level between the upper and lower reservoirs.”*


Expressed in modern terms, Carnot conceived of *temperature* as the *thermal potential*, and temperature differences as *thermal tension*; and he used the term *heat* for the *amount of heat* possessed by a body that is responsible for making it warm and, in the case of a gas, let it expand. This is the *caloric* of the *Caloric Theory of Heat* (for scientific and historical background, see [2,3,4]). Rather than being wrong, Carnot’s imagery relating to waterfalls and caloric is foundational for a unified approach to dynamical macroscopic processes as they are treated in today’s physics of uniform dynamical systems [5] and its generalization to spatially continuous systems ([6,7,8,9,10]; and [5], chapter 11). If we allow for heat to be produced in irreversible processes, we can identify amount of heat, i.e., caloric, with entropy [4,11,12,13,14,15] and formulate its (dynamical) law of balance ([5], chapters 1,4–5; [4], Section 3); and if we take the power of heat—or of water or electricity, for that matter—to denote the rate at which energy is made available in a spontaneous process, we can use the image of the waterfall for constructing a generalized energy principle for macroscopic physical processes.

This outlines the scientific edifice that can be erected upon the Waterfall Analogy. At the same time, it raises questions as to the source of the analogy and our ability to use it for performing scientific work. Answering the latter may very well be an answer to the former as well: Carnot’s Waterfall Analogy is both derivative of and a source of schematic imaginative structures of thought that lend themselves to meaning-making, imaginative structuring, and direct formalization, guided by cognitive tools such as metaphor, analogy, and narrative—this much we can learn from modern cognitive science (see [16,17,18,19,20,21,22,23,24,25,26,27,28]); and if imaginative rationality, operating with the help of the cognitive tools, guides our formal theory construction, it very likely provides us with original images and ideas as well.

### 1.2. Our Starting Point

Summing up, for the present discussion, we simply accept (a) the Caloric Theory of Heat and (b) the Waterfall Analogy and show what they can lead to—in other words, we take Carnot by his word and demonstrate where the combination of (a) and (b) takes us. Furthermore, we argue that (c) Carnot’s and our ability to come up with and use the Caloric Theory and the Waterfall Analogy arises from the power of imaginative rationality (this assumes as well that we are embedded in a culture where this imagery has arisen and is handed down through tradition). We put these assumptions in the context of recent advances in unified presentations of macroscopic physical science [5], which means that (1) we extend the Caloric Theory of Heat by assuming that caloric is produced in irreversible processes, and (2) we add the generalized meaning of power and energy to the imagery suggested by (a) and (b), where the original ideas and their extensions are driven by (c). We consider this model the most productive and generalizable element of Carnot’s legacy, for both theory and applications in macroscopic physics, chemistry, and engineering.

This means that the current paper is less of a historical tale than a sketch and continuation of Carnot’s basic ideas. We recently presented an extensive analysis of Carnot’s memoir *La puissance du feu* [4], where we traced the roots of Carnot’s (and his predecessors’ and contemporaries’) conceptual background to basic forms of human understanding arising in experiencing and imagining ([4], Section 2) and embedded his achievement in the history of thermodynamics from around 1670 to 1865 ([4], Section 3; additional detail on the history relating to Carnot’s work can be found in [29,30,31]). Naturally, Carnot’s Waterfall Analogy played an important role, but we did not dwell on either its origin or its reception, nor on its power for a modern unified approach to macroscopic physical processes in theories of uniform dynamical systems and continuum physics. Here, we want to fill in this gap in the most straightforward manner possible, which, as we shall see, follows from accepting Carnot’s imagery of waterfalls and caloric.

### 1.3. Outline of the Paper

We start with waterfalls, with how we experience them as an agentive phenomenon, and with how our body and mind turn the experience into an extremely simple schematic, i.e., abstract and structure. Learning about this structure and its elements—namely, height of fall, flow of water, and power of the fall—prepares us for transferring the imagery to thermal processes (Section 2). In Section 3, we show how recognizing Heat as a member of the family of causative phenomena, i.e., of the Forces of Nature, lets us understand the Waterfall Analogy and the Caloric Theory as a unified imaginative structure. This allows us to explain heat engines in the simplest manner possible. Moreover, we show how the Carnot cycle is motivated by both the Waterfall Analogy and the Caloric Theory. In Section 3.5, we describe possible origins and the reception of the Waterfall Analogy in the 200 years following the publication of the memoir.

In Section 4, we shall generalize the cognitive foundations of Carnot’s approach by tracing its source to human experiencing. A waterfall can be experienced as the archetypal physical phenomenon, and we owe Carnot for showing us how to make use of the imaginative power it arouses in us. We demonstrate how the Waterfall Analogy arises in and gives form to schematic, metaphoric, and analogical structures of thought. We explain how to distinguish between metaphor and analogy, and so give analogy its proper place in theory construction in macroscopic physical science.

Finally, we show how to extend Carnot’s approach to a generalized energy principle, and how his example of analogical structure mapping [24,25] allows us to create a unified form of models of thermal, hydraulic, electrical, chemical, and mechanical phenomena (Section 5). Letting the same basic imaginative structure arise repeatedly in our mind for creating analogical forms of models across the field of physical phenomena is a testament to the power of what Carnot made possible with his Waterfall Analogy and the Caloric Theory of Heat. Therefore, our treatment of Carnot’s work should be of interest for anyone concerned with the foundations of macroscopic physical science in general and of thermodynamics in particular. In Section 6, we present our summary and conclusion.

## 2. Waterfalls and Gravitational–Hydraulic Processes

Before we outline the Waterfall Analogy in Carnot’s memoir [1] and show how it applies to his treatment of heat engines (Section 3), we sketch what we can learn from schematizing our experience of the flow of water in the gravitational field near the surface of our planet. While the discussion presented here uses a modern form, it parallels Carnot’s assumptions in most of their aspects and so prepares us for applying analogical structure mapping [25] between hydraulic and thermal phenomena.

### 2.1. Schematic Elements of Waterfalls

No two waterfalls are exactly alike, so what characterizes a waterfall? This is where our ability to schematize experience comes in: all waterfalls have a high point from where water falls and a lower point to where it falls; as a corollary, a waterfall is characterized by height or level difference; and, quite obviously, there is a flow of water of a certain magnitude in a waterfall. These two properties, *height* and *flow*, are shared by all waterfalls. Finally, experiencing a waterfall impresses the notion of causal power on us—again, all waterfalls share the property of being powerful; the falling water causes other things to happen—a waterfall is more or less powerful, i.e., it leads to greater or lesser effects.

There are special properties of falls that not all of them need to share, most importantly the fluid flowing—we can imagine falls of fresh water or salt water, of sand or of hydrocarbons like on Saturn’s moon Titan; and the terrain over which the fluid flows or falls—does it actually fall freely (only impeded by air), or does it flow over a (however steeply) sloping terrain so that its flow will be resisted by the terrain?

We should be interested in the question of whether it is possible for the power of a fall of a fluid to depend only on the basic schematic characteristics shared by all falls, i.e., upon the height of fall and flow? What about the type of material falling, and the obstruction of the fall due to the terrain? The former has an important story to tell: there is a schematic/abstract entity behind the falling water that actually characterizes a fall at the surface of our planet. We already know the answer: it is the *weight* (or the gravitational mass) of fluids. The latter introduces us to the notion of *loss of power* and what that entails.

Clearly, a “minimal” (i.e., fully schematic or abstract) waterfall is one where the weight or the mass of water or any other fluid flows in free fall from the upper to the lower level, and where the type of fluid does not matter, i.e., where the “minimal” properties are just two: height of fall and flow of weight of fluid. Note that a given characteristic allows for differentiation: height of fall can be high or low, and flow can be strong or weak. When formalizing our experience, we introduce variables for these two characteristics.

Note that upper and lower levels—especially the upper one—give us a feeling of intensity. If we stand near a very high waterfall, we have to stretch our neck to see the upper level, we might be awestruck by the mere height, and we experience an emotion similar to what we know as psychological tension. Our most direct mechanisms of perception offer us *differences* of qualities or degrees of intensity: differences of height in a landscape, differences of brightness, temperature, pressure, and speed in nature; or differences in health, or price differences of goods in our economy. Differences of intensities or qualities are felt as *tensions* and are said to be driving forces for things to happen.

What we have described here are the characteristics, both general and specific, of a waterfall as a Force of Nature. Experiencing Forces of Nature makes an agentive, causative perceptual unit, a gestalt or figure [4], available to us. We shall discuss this figure at the beginning of Section 3 in the context of Carnot’s treatment of Heat as a Force of Nature.

### 2.2. Power of and Energy Made Available by a Waterfall

As we stand at the bottom of a waterfall, it is quite natural for us to speculate that its power should be proportional to both its height, i.e., the tension established, and the strength of the flow. If we call the factor deriving from the difference of upper and lower levels in the fall of water *gravitational tension* (Figure 1a), we can say that the power of a waterfall will be given by the product of gravitational tension, $∆\varphi_{G}$, and flow (current) of mass of fluid, $I_{m}$:(1)$$P_{grav}=\left(\varphi_{G,high}-\varphi_{G,low}\right)I_{m}$$We disregard questions of signs of potential differences and flows and just think in terms of absolute values. Gravitational tension is the product of strength of gravitational field and height of fall: $∆\varphi_{G}=g∆h$.

We now take a first step towards the notion of energy. Power gives us the feeling of the rate of causation, i.e., how fast something that is caused can arise: how fast the ground at the bottom of the waterfall is eroding, or how fast we can grind grain in a mill powered by the waterfall. If we want to know how much has been caused, how much of the ground has been eroded or how much grain has been ground, we wait for some time and count. We can imagine this causal agent—the falling mass of water—to make something available so the caused process(es) can arise, and we call this something *energy*. We shall say that falling water makes *energy available* ([5,32,33,34,35,36]); it does so over time, and according to the measure of its power:(2)$$E_{av(grav)}=\int_{t1}^{t2}P_{grav}dt$$

Since the height of fall can be taken to be constant, this is equal to the product of gravitational tension and the total amount of mass that falls from top to bottom. The index *av* is used for expressing the idea that energy has been made available by the falling water or, more formally, by the spontaneous gravitational process. Note that $E_{av(grav)}$ is neither the energy of a systems (it is not a storage term, but a concept describing a process), nor is it “gravitational energy” simply because there are no forms of energy [37]—the index just tells us that the quantity of energy calculated in Equation (2) is made available in a gravitational process.

Moreover, despite the step we have taken with Equation (2), we have not yet created a generalized energy principle. As far as the concept of energy is concerned, we are barely a step further than the point where Carnot was in 1824—we understand the notion of power (see Figure 1), but not of energy in its extended sense.

### 2.3. Lifting or Pumping Water

Water can go the other way, from a lower to a higher level, but experience tells us that it does not do this by itself: it must be carried up, or it must be pumped; and, to make this possible, there needs to be an agent—a process powering the lifting of water. This agent could very well be water falling over a cliff; and just as in the case of a waterfall, we can construct the concept of power related to the aspects of vertical distance from bottom to top and current of water pumped—this idea is expressed visually in Figure 1b. Furthermore, it makes sense to construct the mathematical relation for power of a forced process, just as in Equation (1).

If the pumping of water were accomplished with the help of a pump powered by a device such as a water wheel (turbine) powered by falling water, we could picture a chain of processes as in Figure 2. The diagram is obviously incomplete since we are not saying anything about the intervening phenomenon; somehow, the power of the fall of water is “copied” to the power of the process of pumping or raising water. Moreover, the diagram it is unrealistic, since it seems to suggest an ideal case of operation of physical processes (see Section 2.4).

This presents us with another opportunity for expanding the notion of energy: we can interpret the communication between a causing and a caused process as energy being handed from an agent to a patient ([20], p. 80); speaking more formally, *energy is exchanged* between an agent and a patient [4,38]. After all, the falling water has made energy available, so why should it not be used? The idea that coupling is achieved by “handing something” from an agent to a patient [20] makes the concept of power, and therefore energy, even more real—there is more than the mere definition of a quantity, however suggestive, as the product of two factors, as in Equation (1).

### 2.4. The Role of Fluid Medium, Ideal Processes, and Loss of Power

Now, assume that when we use the device suggested in Figure 2, all the water that has dropped will be returned to its original level; therefore, the power of the forced process will be equal to the power of the causing waterfall, i.e., $P_{grav}\left(1\right)=P_{grav}\left(2\right)$, as in Figure 2 (we disregard different signs for the two expressions for power). In other words, the power made available in the fall of water is used or consumed when pumping water back to where it came from. Looked at from outside, it appears as if nothing is happening even though the machinery represented in Figure 2 is running: water running downhill is returned uphill, and power made available is consumed. This sounds like perpetual motion with no effect upon the outside world.

Consider an even weirder case than this. If there were ways of using weight (or mass) that were preferable to letting water flow, i.e., if it were possible, by using maybe sand or crude oil, to make mass produce a greater quantity of power, then it would be sufficient to distract a portion of this power to make mass rise again back to where it came from, to restore things to their primitive state; thereby, this would be not only perpetual motion, but the indefinite creation of motive power without the consumption of mass or any other agent. Such a creation is completely contrary to the ideas received up to now, to the laws of Mechanics and sound Physics; it is inadmissible. *We must therefore conclude that the maximum motive power resulting from the use of water in a waterfall is also the maximum motive power achievable, by any means whatsoever*.

(In case the langue in the previous paragraph strikes the reader as somewhat old and odd, it is a translation of a section of Carnot’s argument that the type of fluid powering a heat engine is of no consequence; we changed it a little bit and replaced caloric by weight, and steam by water, to make the statements fit an engine powered by a waterfall; see [1], pp. 20–22, emphasis in the original.)

Never mind the impossibility of perpetual motion machines making power available without end: the case depicted in Figure 2 does not occur in nature for a different reason. The power of pumping water coupled to a waterfall will always be less than the power of the falling water, i.e., $P_{grav}\left(2\right)<P_{grav}\left(1\right)$. Assuming equality constitutes an imagined limiting case, an imagined ideal operation of physical processes: we can imagine it and work with it in our models, but it will never be achieved in real life. The question is, how do we understand the observation that tells us that $P_{grav}\left(2\right)<P_{grav}\left(1\right)$?

To understand the inequality that is established in real cases, it suffices to consider the driving process, i.e., the waterfall. Clearly, freely falling water is an idealized model. In reality, there will be some air resistance and, even more importantly, resistance if the flowing water will pass partly over some rocks, i.e., if the fall is not properly vertical all the way from top to bottom. This means that there is a part of the fall for which there is no apparent caused process such as lifting some water—this constitutes a “true loss” of power. While $P_{grav}\left(1\right)$ and $E_{av,grav}(1)$ are still the same for given total height of fall and flow, some of the energy made available by this fall does not arrive at the point where the lifting of water is accomplished.

This is even more apparent in the following situation. Imagine that, due to the power of Sunlight, water (vapor) is “pumped” up in the atmosphere to a height higher than the upper level of a waterfall. The water will first fall as rain and then flow downhill over some terrain before coming to the edge of a cliff from where it may fall “freely”: the entire height difference (gravitational tension) from the clouds to the upper edge of the cliff will be “wasted”—no “motion” will have been powered, even though there appears to be a potential, i.e., a level difference, for it.

We have discussed the issue of *loss of power* in anticipation of Carnot’s notion of “true loss” of the “power of a fall of caloric” (see Section 3.2). Note that this discussion extends the list of schematic attributes of flowing water in general or of a waterfall in particular: we should add schemas of obstruction, hinderance, impediment, resistance, etc., to spontaneous processes that, metaphorically speaking, correspond to the downhill flow of some fluidlike stuff. These and the previously introduced basic schemas of levels and height (level differences or tensions), amount, weight, mass, flow, and power are all part of our common cognitive heritage. Carnot’s use of mostly natural everyday language, as revealed in the quote above in Section 1.1 and additional ones we shall encounter in Section 3.2, is a testament to this fact; and as we said above, we will make use of the power of imagery present in Carnot’s language.

## 3. Waterfalls and Heat as Forces of Nature

We start, as Carnot did in his memoir, by embedding the notion of Heat in our experience of Forces of Nature, and briefly outline the basic characteristics of such Forces. Following this, we retrace Carnot’s Waterfall Analogy applied to caloric and heat engines. We briefly sketch the ideas related to ideal processes and explain why the fall of caloric should be independent of the medium used in an engine. We show how the Waterfall Analogy is intimately related to the Caloric Theory, and how this relation motivates the Carnot cycle. Moreover, we discuss Carnot’s quest for a general theory of Heat and conclude by asking where we might find the source for his Waterfall Analogy, and how this idea fared in the 200 years since its publication.

### 3.1. Heat as a Force of Nature in Carnot’s Thermodynamics

These are the opening lines of Sadi Carnot’s book [1], *Réflexions sur la puissance motrice du feu*, of 1824 (pp. 1–2):


*“We are all aware that heat can be the cause of movement, that it even possesses great motive power: the steam engines, now so ubiquitous, are a proof that speaks to anyone who can see.”*



*“It is to heat that we must attribute the great movements which attract our attention here on Earth; it is to heat that we owe the agitations of the atmosphere, the rise of clouds, the fall of rain and other meteors, the currents of water which channel the surface of the globe and of which man has succeeded in using but a small part for his own purposes; finally, earthquakes and volcanic eruptions also recognize heat as their cause.”*



*“It is from this immense reservoir that we can draw the moving force necessary for our needs; nature, by offering us fuel everywhere, has given us the faculty, at all times and in all places, of giving birth to heat and to the power which results from it. To develop this power, to appropriate it to our use, such is the object of heat engines.”*


This is the account of a *Force of Nature* we would call *Heat*. A Force of Nature is a perceptual unit (a gestalt) formed by imagination when we experience causative phenomena. (Note: We do *not* use the term *Force* as is customary in mechanics. In order to discriminate *force* in mechanics from *Force* (as in *Force of Nature*), we shall capitalize the term in the latter case, as we do when we speak of *Heat*, *Electricity*, or *Water*—and others—as Forces.)

Importantly, even though a Force is recognized as an experiential whole, it can be analyzed as having a few characteristics, of which *intensity*, *extension* (“size” or amount), and *power* are the most basic. In the end, we commonly identify a Force through what it causes, i.e., through its power. However, recognition starts with emotion and feeling aroused as a consequence of the perception of differences of qualities and intensities (see Section 2.1). Furthermore, in some cases, we easily notice the extension of a causative phenomenon: the area covered by Rain, the size of a Fire, or the amount of Water. However, there are less obvious cases where we deal with invisible Forces such as Wind, Electricity, and Heat (in [4], Section 2, we presented an account of how experiencing and recognizing the extensive aspect of a Force such as Cold or Heat arises in the imagination).

Typical examples of Forces—where the term Force of Nature is habitually used—are Sunlight, Water, Ice, Fire, Wind, Rain, Gravity, and Substances (including Food and Medicine). We call these *Primary Forces* [38]. Physics has evolved as a handful of theories of what we call *Basic Forces of Nature*: these comprise Fluids, Electricity and Magnetism, Heat, Substances, Gravitation, Rotation, and Linear Motion. In the course of this essay, we shall see how applying the Waterfall Analogy to the list of Basic Forces helps us gain access to a unified representation of these phenomena.

Summing up, if we accept Heat as a member of the family of Forces (of Nature), we should expect it to reveal the same three basic characteristics: intensity (and its differences), extension (size, amount), and power. These would be temperature (and temperature differences, i.e., thermal tensions), amount of heat, and power of heat (or, more formally, thermal power).

### 3.2. What Carnot’s Waterfall Analogy Tells Us About Heat as a Force

In the quote from [1], p. 28, we omitted a footnote. Carnot seems to apologize for explicitly using schemas created in our experience with fluids, their metaphorical projection to Heat, and analogy created in this manner:


*“The subject matter here treated being quite new, we are forced to use expressions which are rather unusual, and which perhaps do not have all the desired clarity.”*


Philosophies of science and the mind have come far enough since then so that we do not have to apologize any longer—this is how we, and scientists and engineers as well, think! We shall take up the issue of schematism, metaphor, and analogy in some more detail in Section 4 (see also [4] for more detail).

We assume that our experience of Water as a Force and the schematic characteristics that are created as a consequence of this experience serve as a source domain of knowledge which is expressed in natural language: water flows, falls, is pumped; it acts, causes, drives, powers; it is acted upon, caused, powered; its flow is hindered, obstructed, etc. Even though all these terms refer to schematic/abstract elements of understanding, they nevertheless speak of direct truths.

We now outline what happens if we use the same schemas for speaking metaphorically about Heat: in a nutshell, it makes the Force of Heat analogous to that of Waterfalls (on metaphor and analogy, see Section 4). In Table 1, we present elements of structures that are mapped from the domain of the Force of Water to the Force of Heat as a summary of what we are going to describe here. With the exception of conservation of caloric, we can accept every single mapping for constructing a generalized approach to theories of macroscopic physics [5], with the Waterfall Analogy taking center stage.

#### 3.2.1. Applying the Waterfall Analogy to the Power of Heat

While the passage quoted in Section 1.1 is the most direct and most beautiful rendering of the Waterfall Analogy in Carnot’s memoir ([1], p. 28), it is by no means the only one where the imagery of falling caloric is expressed. Wherever this appears in the text, it becomes evident that the Waterfall Analogy and the Caloric Theory of Heat form a tight whole—we really cannot separate one from the other. We shall discuss this point in more detail below in Section 3.2.7.

We have already claimed that we shall take Carnot by his word and immediately proceed to an expression for thermal power which is the analog of Equation (1); clearly, then, the (thermal) power of a fall of caloric must be proportional to both the thermal tension $∆T=T_{high}-T_{low}$ and the flow of caloric:(3)$$P_{th}=\left(T_{high}-T_{low}\right)I_{C}$$Here, $P_{th}$ is the power of caloric falling through a temperature difference ∆T, and $I_{C}$ denotes the current of caloric.

A word of caution and fairness is in order at this point. Stating the relation for the power of a fall of caloric (Equation (3)) as if it were Carnot’s is not totally fair to him, since, after describing the Waterfall Analogy (Section 1.1), he advised caution as to how the (motive) power of a fall or caloric would depend upon the upper and lower temperatures in the heat engine ([1], pp. 28–29):


*“In the fall of caloric, the motive power undoubtedly increases with the difference in temperature between the hot and cold bodies; but we do not know if it is proportional to this difference.”*


Not assuming the simple form from direct analogical mapping meant that Carnot needed to prove the point. We showed how he did this [4] and how the proof can be cast in more modern language (see [5], chapters 4 and 10), so we do not need to repeat this here. After all, our aim is to demonstrate how accepting the Waterfall Analogy as directly and fully as possible helps us create a unified representation of macroscopic physical systems and processes.

Summing up, if we let our understanding be guided by the archetypal structure of a waterfall, we should never confuse a quantity of heat falling with the power of its fall. After all, no one will ever confuse a quantity of water falling with the power of its fall. It is then clear that we should never come up with the idea that heat is power (or rather, energy), and that in its fall, part of it will be converted into motive power while the rest must be discarded to the environment.

#### 3.2.2. The Loss of Power in a Fall of Caloric, and the Production of Caloric

The mechanical power of a real heat engines comes nowhere close to what the Waterfall Analogy suggests, i.e., by what we should expect from Equation (3). After all, the power of Heat calculated by this equation is that of an ideal fall of caloric from $T_{high}$ to $T_{low}$, and nothing is ideal in the real world. What is important is how Carnot explains the difference ([1], p. 12 and p. 23):


*“Wherever there is a difference in temperature, wherever there can be a restoration of equilibrium in the caloric, there can also be production of motive power.”*



*“Since any restoration of equilibrium in the caloric can be the cause of the production of motive power, any restoration of equilibrium without the production of this power must be considered as a real loss.”*


We need to explain briefly what calorist like Carnot meant by restauration or re-establishment of equilibrium in caloric. This has nothing, per se, to do with equilibration in and between bodies. It is Carnot’s expression for caloric passing from a hot to a cold body, to where it originally came from. For a calorist, caloric was not produced in the furnace at high temperature, but “evolved”. By chemical action or friction, latent heat was brought out to become sensible; this sensible heat was passed through the heat engine to end up, in the form of “latent heat”, back in the cold environment. So, *“any restoration of equilibrium in the caloric”* simply means any fall of caloric through a temperature difference.

The first part of the quote reiterates the idea expressed in the waterfall imagery: whenever caloric falls through a temperature difference, a certain power is associated with this which, potentially, drives Motion. The keyword here is “potentially”, because, as the second part of the quote tells us, caloric can fall without producing Motion, and this constitutes a “real loss”.

As we see from our description of how to speak about and understand waterfalls, this is perfectly analogous to the loss of “motive power” in real waterfalls. Carnot’s insight is quite important for practice, since it tells engineers where to look for inefficiencies in engine design, at least as far as the transport of caloric from the furnace to the cooler is concerned. Moreover, the insight provides the motivation for the structure of the Carnot cycle (see Section 3.3).

But what happens when there is a “true loss” of power? Apparently, the caloric falling through a certain temperature difference does not do what it could potentially do or what we would like it to do, namely, power Motion in an engine. At first, it appears as if nothing happens upon the *“useless restauration of equilibrium in the caloric”*. However, consulting waterfalls and the flow of water in general (Section 2), or the host of phenomena where other fluidlike quantities such as electric charge or momentum flow “uselessly”, i.e., without “motive power” being produced, we know that there is always something that happens: a thermal process is caused, which can mean only one thing—caloric is made to appear where there was none before (calorists would have said that caloric has been evolved, i.e., latent caloric has been “converted” into sensible caloric). This means that we will take a step the calorists were loath to take: we will accept that *caloric is produced* in irreversible processes; and we do this not just for when we rub our hands, let charge flow through a thin wire, or burn a fuel, but also when caloric uselessly falls through a temperature difference!

The last of these processes, the conduction of heat—when heat (caloric) flowing produces more heat—caused great difficulties in the time after Carnot and leading up to W. Thomson’s and R. Clausius’ work; it still does so today when it is so clear, as it was to Carnot, that the main irreversibility in heat engines comes from the necessary transfer of heat into and out of the working fluids ([33,39,40]; [5], chapter 9). Here is what W. Thomson said about the challenge of loss of power in the flow of heat, as late as 1849 ([41], footnote on p. 545):


*“When ‘thermal agency’ is thus spent in conducting heat through a solid, what becomes of the mechanical effect which it might produce? Nothing can be lost in the operations of nature—no energy can be destroyed. What effect then is produced in place of the mechanical effect which is lost?”*


The waterfall and the loss of power made some additional history in relation to thermodynamics when, reportedly, William Thomson and J. P. Joule met in the Alps when Joule was there on his honeymoon in 1847 [42,43]. According to the story, they endeavored to measure an anticipated rise of temperature of the water arriving at the bottom of a waterfall after having fallen “uselessly”. We may take this as one of the many examples of “true losses” occurring in physical processes for which interest was inspired by Carnot’s use of waterfall imagery. Apparently, Joule and Thomson were unsuccessful, but the question of loss, and how to interpret what it meant, kept coming up. What we know about Joule suggests that this would have been another example for him showing that the Caloric Theory had to be wrong; Thomson, on the other hand, was not yet ready to give up on it, but like all calorists, he stubbornly held on to the all-to-literal interpretation of heat (caloric) as a type of (imponderable) substance (see [41], footnote on p. 545; [3], chapter 7).

#### 3.2.3. Perpetual Motion and the Medium Used in Heat Engines

Carnot sought to argue that the expression for the power of a fall of caloric will be universal, i.e., independent of the fluid used for transporting caloric from a furnace to a cooler. He used this reasoning involving combining an ideal heat engine with an ideal heat pump, assuming that if there were a means of using caloric that was superior to another, then additional power would be produced, which would result in perpetual production of motive power. Here is the original statement which we already made use of in Section 2.4 ([1], p. 20–22, emphasis in the original):


*“Now, if there were means of employing heat that were preferable to those we have used, i.e., if it were possible, by any method, to make caloric produce a greater quantity of motive power than we have done by our first series of operations, it would suffice to distract a portion of this power to make caloric rise, by the method just indicated, from body B to body A, from the cooler to the furnace, to restore things to their primitive state, and thus be in a position, and thus be in a position to recommence an operation entirely similar to the first, and so on. This would be not only perpetual motion, but an indefinite creation of motive force without the consumption of caloric or any other agent. Such a creation is completely contrary to the ideas received up to now, to the laws of Mechanics and sound Physics; it is inadmissible (footnote). We must therefore conclude that the maximum motive power resulting from the use of steam is also the maximum motive power achievable by any means whatsoever.”*


After considering a simple gas in place of steam, and after outlining what we today call the Carnot cycle (see Section 3.3), Carnot repeats his important conclusion that the power of a fall of caloric must be independent of the working fluid used in a heat engine ([1], p. 38, emphasis in the original):


*“This leads us to the following general proposition:”*



*“The motive power of heat is independent of the agents used to produce it; its quantity is determined solely by the temperatures of the bodies between which the heat is ultimately transported.”*



*“The implication here is that each method of developing motive power achieves the perfection to which it is susceptible.”*


Needing to reason with perpetual motion for arriving at what appears to be a simple result sounds somewhat clumsy, but we must remember that Carnot needed to prove the validity of what would follow directly from the Waterfall Analogy and the Caloric Theory. We are in a much simpler position today: we know that what is true for the abstract notion of a waterfall is equally true for a fall of caloric. We not only have two hundred years of detailed experimental evidence telling us that the analogy works, but we also understand how to wield metaphorical and analogical thought for which we do not have to apologize: we know we can accept a degree of imagistic abstraction not yet available to the researchers 200 years ago. If a schematic and imaginative quantity such as mass or caloric falls from a point of potential 1 to a point of potential 2, the power of the fall is invariably given by Equation (1) or Equation (3), and these expressions apply independently of the means by which mass or caloric are transported and what happens after energy has been made available (i.e., if what follows is an ideal or a partly or fully dissipative process).

#### 3.2.4. A Note on Motive Power and on Power in General

Our discussion of the power of falling and lifting water (Section 2.3) suggests that measures of power should be associated with both causing and caused processes, and that we should distinguish between these measures because they might be different. We do not see this distinction made by Carnot; in the footnote on [1], pp. 6–7, he writes:


*“We use the term motive power here to designate the useful effect that a motor is capable of producing. This effect can always be likened to the lifting of a weight to a certain height; as we know, its measure is the product of the weight multiplied by the height to which it is supposed to be lifted.”*


This means that *motive power of Heat* is $P\left(2\right)$ in Figure 2 (or P in Figure 1b). It is not clear to us whether, for Carnot, the *motive* power of Heat is the only power that “exists” (i.e., that there is no independent *power of Heat*), or if power of Heat, $P_{th}$—i.e., what we determine from Equation (3)—is simply *identical* to *motive* power. However, this then leaves us with the question of how to imagine *loss* of motive power.

If we do explicitly distinguish between the *power of a fall* and the *power associated with the process of lifting* or causing, if we distinguish between *energy made available* and *energy being used*, then we can avoid some confusion. First of all, we can focus on what happens when two Forces such as Heat and Motion interact; and second, we can make clear that the power of a fall of caloric will always be calculated by Equation (3), no matter what will follow the fall, whether it will be ideal, unimpeded, or real, i.e., impeded in some way.

Accepting that Equation (3) applies under all and any circumstances, and that it is an example of what we mean by the power of an agent going through a spontaneous process and energy being made available, will be instrumental for creating a unified representation of physical processes (see Section 5)—Carnot’s power of Heat, derived from his waterfall image, will serve as the blueprint for analogous cases in Fluids, Electricity, Substances, and Motion. It helps explain the notion of loss of power as something that happens in the interaction of two forces, an agent and a patient: the agent makes a certain amount of energy available according to what we have formulated in Equations (1)–(3), and it is some of this available energy that is not used by the caused process such as Motion in a heat engine. However, the energy not used is not lost: it is used for producing caloric. We now proceed to create visual representations of the ideas of power of a fall, power of lifting, and “loss” of power as they emerge in ideal and dissipative heat engines.

#### 3.2.5. Process Diagrams of Ideal and Real (Dissipative) Heat Engines

We can summarize what Carnot’s Waterfall Analogy teaches us in graphical form, where we use visual metaphors for what the idea of Water as a Force, such as in a waterfall, suggests to Carnot and to us. We continue the diagrammatic representation we started in Figure 2; see Figure 3.

In process diagrams, we show the interaction of Forces, i.e., of an agent and, typically, two patients, plus the exchange of energy between them. In both ideal and real heat engines, caloric flows in at high temperature and out again at low temperature. The power of the process is calculated with the help of Equation (3), and it will be the same for both cases (Figure 3a,b). As a consequence of the coupling, all or part of the energy made available, which we calculate according to Equation (2), is used by the patient; we can imagine the patient to be either Rotation or Electricity. The extensive quantity of the caused Force flows from low to high potential, i.e., it is pumped.

The difference between ideal and real engines is found in how much of the energy made available by the fall of caloric ends up “in the hands” of the desired process. If it is 100%, i.e., if the efficiency of the engine equals 1, then the coupling is ideal. In a real engine (Figure 3b), a part of the energy will be used by the dissipative phenomenon where caloric is produced. The caloric generated in the engine will leave at $T_{low}$ together with the caloric that took its fall from $T_{high}$ to $T_{low}$. Importantly, producing caloric is analogous to pumping it in a heat pump, where the lower temperature is absolute zero, i.e., 0 K, and the upper temperature is the temperature of the material in which dissipation takes place. Therefore, the general relation for the production rate of caloric $\Pi_{C}$ in a dissipative process is ([5], chapter 4.4):(4)$$\Pi_{C}=\frac{P_{diss}}{T}$$

Note that, in general, dissipation will happen over a range of temperatures. This means that, in an engine, caloric will be produced at rates that are initially lower than what is indicated by the additional amount flowing out at $T_{low}$. There will be additional caloric produced due to its “useless fall” through the materials of the heat engine, from whatever temperature it is first produced at all the way down to $T_{low}$.

#### 3.2.6. A Note Concerning Different Measures of Efficiency

In the paragraph leading up to Equation (4), we stated that the efficiency of an ideal heat engine, one that makes use of the ideal fall of caloric as in an ideal waterfall, equals 1 (or 100%). In a real engine, this value is less than 1.

The measure of efficiency used here is based upon the comparison of available power of a fall of caloric according to Equation (3) and the power related to the primary caused process represented in Figure 3, i.e.,
(5)$$\eta =\frac{P_{X}}{P_{th}}$$

This is how we commonly define the efficiency of coupling of Forces, as in a hydraulic power plant or an electric motor. It tells us the fraction of the energy made available that is picked up by the primary caused phenomenon [5]. In thermodynamics, it has been called second law efficiency.

However, we might be interested in how much of the energy made available in the burning of coal will go toward the mechanical process powered by a steam engine; this certainly was one of Carnot’s questions. Interestingly, the efficiency so defined is less than 1 even for the ideal heat engine represented in Figure 3a. This measure of efficiency has been called first law efficiency or (thermal) energy efficiency. We shall see how to calculate it once we have extended the energy principle sufficiently (see Section 5).

#### 3.2.7. How the Waterfall Analogy and the Caloric Theory Form a Unit

The quote presented in Section 1 ([1], p. 28) shows how to think about caloric if we accept the Waterfall Analogy. Caloric is a fluidlike quantity, it flows and falls, and it can be pumped ([1], pp. 16–20). For falling and lifting to make sense, there need to be intensities that are metaphorized as (vertical) levels or potentials; this is how we understand the concept of hotness and the role of temperature and thermal tension [5,44]. Naturally, there is more to caloric: it can flow into and out of materials and it can be stored in them, and materials such as a simple gas responds to caloric by changes of temperature and volume. However, for now, it suffices to accept what we can learn about caloric if we consider the Waterfall Analogy.

Now, consider the inverse argument: if we accept the Caloric Theory, what can we learn about the origin and utility of the Waterfall Analogy? We believe here lies much of the genius of Carnot: his predecessors and contemporary colleagues used the Caloric Theory for studying the response of materials to amount of heat (i.e., caloric) possessed by them. No one took the leap and applied the schematic aspects of caloric to create the image of its *fall through a temperature difference*. If we imagine caloric residing in bodies and flowing in and out—in other words, if we accept the metaphor of a fluidlike quantity—we should be allowed to compare it to water and what we know happens when water falls over a cliff. Here are a few quotes from Carnot’s memoir where the role of caloric hints at how it acts when it falls:


*“The production of motive power is thus due, in steam engines, not to an actual consumption of caloric, but to its transport from a hot body to a cold body, …”*

*([1], pp. 10–11; emphasis in the original).*



*“[…] the motive power of the heat also depends […] on what could be called, in effect, the height of its fall […]”*

*([1], p. 28)*



*“In other words, the motive power produced would be exactly proportional to the drop in caloric.”*

*([1], p. 79)*


With this, Carnot concludes the dual move inherent in analogy: he compares water to caloric, and caloric to water; or, more generally, he compares Water as a Force to Heat as a Force, and vice versa (analogy is a bi-directional mapping; source and target domains can switch their roles; see Section 4). Nobody before him seems to have performed this comparison, and not all that many seem to have used the power of this analogy after 1824 (see Section 3.5).

### 3.3. Waterfalls and Ideal Carnot Heat Engines—The Carnot Cycle

By now, we have seen how Carnot thinks of caloric, its fall through a temperature difference, the “production” of motive power, and loss of power. Moreover, we saw how he thought of the maximum “power being developed” and how this depended solely on the fall of caloric from $T_{high}$ to $T_{low}$ and not on the type of fluid used for transporting caloric. The ideal case is obtained if no fall of caloric through a temperature difference is ever squandered. The question then arises how one could achieve this ideal circumstance when employing a material fluid in a heat engine.

So far, we have not used the Carnot cycle, i.e., Carnot’s model of the working fluid in a heat engine undergoing a cyclical process, which allows for caloric being transported from $T_{high}$ to $T_{low}$ in ideal manner (today, we would say, without dissipation occurring). Despite its limitation to heat engines that use a fluid as a working medium, the model can help us clarify a couple of points about the role played by the Waterfall Analogy for understanding how a classical heat engine functions. In particular, we shall learn how to construct a model in which all of the energy made available in the fall of caloric from $T_{high}$ to $T_{low}$ can be made use of for the process of powering Motion—we call the model an *Ideal Carnot Heat Engine* ([5], p. 138; [4,45]).

#### 3.3.1. The Carnot Cycle

Imagine what a mechanism accepting water supplied at a high level $h_{1}$ and discarding it at a lower level $h_{2}$ might look like. We could build a mechanical device where a container can be lowered from $h_{1}$ to $h_{2}$ and then raised back again. We let water fill the container at a constant level $h_{1}$, disconnect it from the source of water, and let it fall to $h_{2}$, while powering a mechanism. Then, the container is drained at $h_{2}$ whereupon it is lifted, now being empty, to $h_{1}$. After this, we start the cycle again, and so on. We do need energy for lifting the container, but since it is empty, this will be less than was made available in the fall of the full container—the machine “develops positive motive power” as a consequence of using the ideal fall of water from $h_{1}$ to $h_{2}$.

Now, map this mechanism to a heat engine employing a simple gas such as air as the working fluid inside a cylinder fitted with a piston. The air, being initially at temperature $T_{high}$, is brought in contact with the furnace (body A in the first quote in Section 3.2.3; see [1], pp. 20–22) which is also kept at $T_{high}$. Carnot imagines that there are materials separating the working fluid from the furnace that allow for “caloric to pass easily”, i.e., without needing a temperature difference for caloric to flow from the furnace into the fluid. In Carnot’s words ([1], pp. 32–33):


*“1. Contact of body A with the air contained in the cylinder, or with the wall of this cylinder, a wall which we will assume transmits caloric easily. This contact brings the air to the same temperature as body A […].”*



*“2. The piston gradually rises to [a new position]. Contact is always maintained between body A and the air, which is kept at a constant temperature during rarefaction. Body A provides the caloric needed to maintain constant temperature.”*


In step 2, he describes isothermal expansion of the gas which requires the right amount of caloric to be supplied to the gas (this is the filling stage in the hydraulic analog described above). Then, the cylinder is removed from contact with the furnace; the gas is allowed to expand adiabatically to the point where its temperature has fallen to $T_{low}$. The cylinder is then brought in contact with the cooler (which is maintained at $T_{low}$ as well); the gas is compressed isothermally while the caloric taken up before is transmitted “easily” to the cooler (Body B). Once all the caloric added through heating has been ejected during the cooling operation, the cylinder is removed from the cooler and the gas is compressed adiabatically until the temperature $T_{high}$ has been reached once again. The cycle is complete and can recommence.

This is the description of the Carnot cycle which allows for ideal (non-dissipative) operation, since


*“This condition will be fulfilled if, as we noted above, there is no change of temperature in the bodies that is not due to a change of volume, or, which is the same thing expressed differently, if there is never any contact between bodies of significantly different temperatures.”*

*([1], p. 38)*


#### 3.3.2. Ideal Carnot Heat Engines

We can now formulate the model of *Ideal Carnot Heat Engines* (ICHEs). Assuming that we can neglect mechanical (or electrical or other) irreversibilities, an ICHE is a thermal engine that is heated (receives caloric from the furnace) at constant upper temperature $T_{high}$ and cooled (emits caloric to the cooler) at constant lower temperature $T_{low}$; there shall be no heating or cooling at temperatures anywhere between $T_{high}$ and $T_{low}$. Importantly, this is the thermal engine that achieves maximum power. (We can create models of ideal Diesel or other cycles; however, since heating and cooling take place at a range of temperatures between $T_{high}$ and $T_{low}$, such cycles do not achieve maximum power even though they operate reversibly.)

An engine going through (Carnot) cycles operates intermittently, and so does our model of a hydraulic engine described above, which we used for motivating the analogy between hydraulic (gravitational) and heat engines. However, a turbine powered by a waterfall operates continuously. We shall use this strong version of the Waterfall Analogy and imagine heat engines in general, and ICHEs in particular, that can be operated in continuous mode as well. This has already been suggested in the process diagrams of heat engines in Figure 3. We shall extend this imagery to all manner of processes (to the interactions of different types of Forces of Nature) when we extend Carnot’s suggestions to the fields treated in macroscopic physics (see Section 5).

### 3.4. Carnot’s Quest for Creating a General Theory of Heat

We commented on the fact that Carnot’s predecessors and contemporaries used the Caloric Theory for investigating the effect of caloric upon materials as they undergo thermal processes; an important example here is Laplace’s work on the speed of sound in air and adiabatic processes ([46]; [4], Section 3). Carnot felt that all these investigations did not amount to a general theory of heat—we can assume that he formed this opinion for the simple reason that explanations for heat engines had not been obtained; the theory obviously needed to be extended. Carnot writes ([1], p. 8):


*“To consider the principle of the production of motion by heat in all its generality, it must be conceived independently of any particular mechanism or agent; reasoning must be established that applies not only to steam engines [footnote], but to every conceivable heat engine, whatever the substance used and however it is acted upon.*



*“Machines that do not receive their motion from heat, those powered by human or animal forces, waterfalls, air currents, etc., can be studied in great detail by the mechanical theory. All cases are foreseen, and all conceivable movements are subject to general principles that are solidly established and applicable in all circumstances. This is the hallmark of a complete theory. Such a theory is obviously lacking for heat engines. It will only be available when the laws of physics are sufficiently extended and generalized to make known in advance all the effects of heat acting in a given way on any body.”*


Carnot is certainly correct that heat engines will be a key to such extension and generalization, and here the Waterfall Analogy is a most important building block. We have seen what it leads to most directly, namely the expression for the power of the flow of caloric, i.e., its fall, from a hot body to a colder one (see Equation (3)). This provides us with the most general relation between the three basic aspects of Heat as a Force of Nature (Section 3.1), and it is immediately instrumental in deriving certain constitutive relations for materials, such as gases, that are used in Ideal Carnot Heat Engines (Section 3.3.2; see [4,5]). In this manner, the Waterfall Analogy strengthens the foundations of the Caloric Theory, even before the important step could be taken, much later, where we accept that caloric will be produced in irreversible processes.

Nevertheless, Carnot’s theory of the power of Heat is only one of the building blocks needed for a complete and modern theory of the dynamics of Heat. We need to extend the Caloric Theory (Section 3.2.2) and formulate the law of balance for caloric including a non-negative production rate, relate the production of caloric to “lost power”, create a generalized energy principle (Section 5.2), and develop dynamical models of thermal processes featuring initial value problems [5,8,9,10]. Once achieved, it will serve as motivation for analogical transfer to other physical phenomena (see Section 5).

### 3.5. History of the Waterfall Analogy

We claim that the Waterfall Analogy—augmented by the Caloric Theory—is a powerful tool for scientific work; it certainly was so for Carnot. Let us quickly describe where the tool may have come from, and then discuss what was carried out with and to it in the 200 years since it was formulated.

#### 3.5.1. On the Origin of the Waterfall Imagery in Carnot’s Work

Where does the idea of comparing the operation of a steam engine to a waterfall come from? Certainly, Carnot was embedded in a culture of engineering and scientific practice ([47]; for a synopsis, see [48]). We can find many sources discussing the background Sadi Carnot found himself in, particularly that laid by his father, Lazare Carnot, who worked extensively on mechanics in general and hydraulic machines in particular ([49]; for a wide-ranging discussion of the work of the two Carnots, see [30]). Apparently, Sadi Carnot visited his father who had been exiled to Poland in 1815. According to legend, father and son visited the Warsaw waterworks which had one of Watt’s steam engines working [50]—here, we have a source of direct experience of the interaction of the Forces of Water and Heat that may have inspired Sadi Carnot to explicitly formulate his analogy.

Already in 1868, Saint-Robert drew attention to the analogy between the fall of water in hydraulic engines and the fall of caloric (see [48], p. 85). Norton [31] suggests the analogy was inspired by Lazare Carnot’s work on machine efficiency, including hydraulic machines. Pisano et al. [51] state that Carnot developed the idea from his father’s earlier work on mechanical machines and the analogy between falling water and falling weights. Gilliespie and Pisano [30] imply a connection between Lazare’s work on hydraulics and Sadi’s thinking.

Fox [47] suggests inspiration from contemporary engineering practices, particularly water–power technology. Müller [52] states that Carnot likely developed the idea from his familiarity with waterwheels and hydraulic machines. Lervig [53] mentions the analogy appearing in Clément’s lecture notes, which Carnot likely attended at the Ecole Polytechnique. Grinevald [54] notes influence from hydraulic engineering concepts prevalent in the 18th century.

Whatever the direct source may have been for Sadi Carnot, there is still a more general one not commonly discussed in the history of science: our common cognitive heritage that makes certain forms of experiencing a foundation of understanding, both in individuals and in cultural practice. There is reason to accept waterfalls as an archetype of human meaning making, and its use in analogical reasoning a natural element of our cognition (see Section 4).

#### 3.5.2. The Waterfall Analogy in Thermodynamics Shortly After Carnot

We briefly discuss the reaction of the scientific and engineering communities to Carnot’s work. We can be brief since, as far as the Waterfall Analogy is concerned, there really is not much to tell. Kerker [55] points to a wave of disregard by the engineers soon after 1824. The analogy did not fare much better among those who would later form our traditional version of thermodynamics, chiefly W. Thomson and R. Clausius. Here, we shall mention Clapeyron, W. Thomson and his brother, and Clausius to learn about the impact of the Waterfall Analogy, or lack thereof.

Clapeyron [56] was pretty much the first to take up Carnot’s work some 10 years later. He treated the ideas, again based upon the Caloric Theory, more briefly and more formally; beyond that, what he presented is simply Carnot’s theory—however, without ever mentioning waterfalls nor caloric *falling* through a temperature difference. While the imagery must have been there, being silent about it so soon after 1824 should already warn us that many of those who came after Carnot do not seem to have seen much practical utility in metaphor and/or analogy; it appears—as is the case today—that, for those who pride themselves on their rigorous scientific work and attitude, figurative thought cannot be part of science—it is not objective/literal or formal enough. However, we shall see in Section 4 in what sense schematism/abstraction, metaphor, and analogy form a foundation for formal scientific thought (see also [23,24,27,28,57,58]).

William and James Thomson are somewhat different cases. In W. Thomson (later Baron Kelvin), the Caloric Theory experiences a “bittersweet Indian Summer” [3]. As demonstrated in quite some detail in [59], it was most likely the influence of William’s brother James, an engineer, that kept interest in the Waterfall Analogy and the Caloric Theory alive. At any rate, in 1849, W. Thomson made ample use of the imagery of the fall of caloric (see, for example, [41], p. 569):


*“In the use of water-wheels for motive power, the economy of the engine depends not only upon the excellence of its adaptation for actually transmitting any given quantity of water through it, and producing the equivalent of work, but upon turning to account the entire available fall; so, as we are taught by Carnot, the object of a thermodynamic engine is to economize in the best possible way the transference of all the heat evolved, from bodies at the temperature of the source, to bodies at the lowest temperature at which the heat can be discharged. With reference then to any engine of the kind, there will be two points to be considered. (1) The extent of the fall utilised. (2) The economy of the engine, with the fall which it actually uses.”*


As we know, Thomson gave up on the Caloric Theory in 1851 [60]. In the work of Rudolf Clausius published in 1850 [61], we do not find any mention of caloric or (water)falls any longer, despite the fact that he, like all the pioneers of thermodynamics, made use of Carnot’s basic insight that a heat engine worked by drawing power from the transfer of heat from a furnace to a cooler [3]. The imagery of a *fluidlike quantity* (caloric) *falling* from a higher to a lower thermal *level* had died in physics.

#### 3.5.3. Waterfall Analogy and the Caloric Theory in More Recent Work

The Waterfall Analogy is known fairly well, and we can find mention of it in quite a few sources. Most commonly, though, mention is brief, and authors make short shrift of it since it is deemed to lead us down the wrong path—after all, it is related to the Caloric Theory which is assumed to be wrong; however, it is possible to find some works where the Waterfall Analogy is used for explaining Carnot’s work. Chief among these is [48] (pp. 77–84), where we find a lengthy and quite fair explication of Carnot’s ideas, waterfall imagery and caloric included.

There are some sources where the Waterfall Analogy comes in disguise. If use is made of entropy as a primitive quantity, it is possible to state the work carried out in an ideal Carnot cycle as the product of the difference of high and low temperatures and the entropy communicated to the working fluid in heating (which is equal to the entropy withdrawn again in cooling); see [62] (p. 119). Moreover, in a book on chemical engineering [63], we find a derivation of the efficiency of an ideal Carnot engine, rendered in the form of an electrical flow diagram, where the fall of entropy through a temperature difference is applied as motivation.

Let us see whether looking at how the Caloric Theory faired after 1850 can tell us more about how scientists thought about the Waterfall Analogy—after all, the two are intimately related (Section 3.2.7). Despite the generally held assumption that the Caloric Theory had to be wrong, there were a few authors who, in the last 100 years or so, attempted to bring back, or at least cast in somewhat more positive terms, what Carnot and others before him had accepted about the nature of the extensive aspect of heat, i.e., caloric [11,12,14,15,64]. The authors’ motivation has been to make clear that an Extended Caloric Theory that allows for production of caloric, makes caloric equivalent to entropy in macroscopic thermodynamics—this is expected to greatly help in creating meaning and understanding a science that is often deemed opaque.

Even here, we do not see much mention, and even much less use made of, the Waterfall Analogy. Only in [11,12,14,15,64] do we find the waterfall mentioned: the authors show how, accepting entropy as primitive, it is easy to express the energy released by a fall of entropy/caloric; in [12,14], this is performed only in “quasistatic” form, i.e., with infinitesimals, which basically disregards the overall appearance of a waterfall.

With the exception of Job [12], nobody extends the use of the Waterfall Analogy to other than thermal processes; he discusses the analogy between water mills, heat engines, and electric motors (p. 20), and suggests that the “useless” fall of water in a waterfall is analogous to electrical charge flowing through a resistor (p. 26).

Moreover, there are countless suggestions that Carnot was the inventor of the second law of thermodynamics [65,66,67,68], including the famous treatment of the matter by Feynman [69]. However, these references to the history of the second law usually do not add to our understanding of entropy and its fall through a temperature difference.

Our survey is certainly not complete, but it is an indication of a rather disheartening state of affairs—as a tool for creating models and structuring theories, the Waterfall Analogy has largely vanished from physics. Nevertheless, there is hope for it: we shall demonstrate that much more constructive use can be made of Carnot’s waterfall imagery (see Section 5 and [5,58,70]).

## 4. Archetypes, Schematism, Metaphor, and Analogy

Here, we briefly describe the idea of a Waterfall as an archetype of human understanding and reasoning. Then, we discuss the origins and use of image schemas, their metaphorical projection onto target domains, and how the subsequent emergence of analogy can be understood. This serves to show that metaphorical and analogical forms of reasoning are common everyday sources of our understanding, which may very well underlie the application of schematic/abstract imagery and analogy in Carnot’s Waterfall Analogy. At any rate, metaphor and analogy have been shown to be integral parts of reasoning in physical science [23,24,25,57,71,72], and they have been since its beginning.

### 4.1. A Waterfall as an Archetypal Causative Entity in Nature

An archetype is an experiential whole, a unit or entity appearing in imagination; it is a gestalt or figure [73,74] like so many other perceptual units, yet it is special in that it stands out in a group of similar gestalts. A Waterfall is a Force of Nature, a causative, agentive figure, and there are many more of this general kind. So, what makes it special in this group, what makes it the archetype of the causative entities we encounter in physical processes?

As we see it, the schematic characteristics we apply very generally to all Forces of Nature are largely drawn from images of fluids flowing at the surface of our planet; obviously, they are inherent in a Waterfall as well. Expressed differently, a Waterfall as a Force provides some of the very images which build the foundation of our understanding of Forces in general. All this makes the gestalt of the Waterfall uniquely powerful: it makes it the first among peers, the one from which we learn to understand the others—it is the archetypal character we use for anchoring the analogical mapping between different Forces of Nature (see Section 4.3).

### 4.2. Experiencing, Image Schemas, and Primary Metaphors

We begin where experiencing begins—with the interactions of humans with their natural, social, and cultural environments [38,75]. Through the schematizing action of our mind, these interactions lead to the creation of important image schemas, i.e., basic gestalts having simple schematic structure which makes imaginative reasoning possible [16,76]. Examples of such schemas are scale, vertical level, substance, flow, speed, letting, obstruction, causation, and many more. In cognitive linguistics, they are called image schemas.

These schemas can be projected metaphorically onto target domains (Figure 4a), which may belong to experiential domains stemming from totally different areas (usually those for which we do not have the same simple and direct experiential access that leads to image schemas in the first place). Metaphors of this type are called primary metaphors [21,77,78,79]. A simple example for this is the projection of a combination of scale, vertical level, and speed onto our experience of hotness—witness linguistic metaphorical expressions such as “the temperature is very low”, or “the temperature is rising fast”.

It turns out that much of our metaphorical production, especially for communicating about macroscopic physical processes, emerges from primary metaphors that use schemas derived from the experience of fluids, among which water is the most important. This becomes clear when we realize a preponderance of schemas related to flowing, containment, obstruction, rising and falling, power and causation, etc.

We can now understand how typical Forces of Nature such as Wind or Heat are structured with the help of metaphors. We have described Forces as having three basic aspects—namely, intensity, extension, and power—plus a number of special characteristics (Section 3.1). Wind and Heat are both intense (intensity is high or low, rises or falls). Wind has a spatial extension (blowing through an imagined vertical area) and Heat is characterized by an amount; both Wind and Heat are imagined to flow. In both cases, flows can be obstructed, and both are powerful phenomena. For all these experiential aspects, we have simple everyday metaphorical expressions that are witness to how we understand and think about such phenomena [4]. Importantly, image schemas that are projected metaphorically have enough schematic/abstract structure which lends itself to logical and formal rendering, if we so desire—this is how we can construct mathematical expressions for our imaginative structures [71,80].

### 4.3. Blending as Compression over Phenomena: Origins of Analogy

We can now sketch how *analogies* between our descriptions of different Forces—fluid, thermal, electrical, chemical, etc.—arise. We use a methodology developed in the model of conceptual integration ([81,82,83]) and show how a blend of figurative renderings of fluids and heat can emerge (Figure 4b). In blending theory (conceptual integration theory), we describe knowledge structures as spaces that interact and can be merged. A minimal conceptual integration network consists of two input spaces (experiential domains, such as Fluids and Heat), a generic space (of generic mental structures), and a blended space.

As we just described, experience of fluids and heat supports the formation of, or makes use of, already existing (image) schemas; they populate a *generic space*. These abstractions are projected onto two *input spaces* representing fluids and heat, respectively; this leads to metaphoric expressions of the kind we have analyzed in this section. By blending the experience, we have with fluids and heat, a *blended space* arises in which heat behaves like a fluid that can be generated. We literally learn to “see” heat as a fluid substance concentrated more or less densely in materials, letting them appear warm or cold, flowing spontaneously from hotter to colder places, and being pumped or generated in non-spontaneous processes.

The process in which we create a blend is called *compression*. “The blend is a compression of the input spaces, while the generic space is an abstraction over them” ([83], p. 98); “[a]dditionally, the blend develops emergent properties that are not possessed by any of the input views” (p. 96). Summing up, through the act of compression of different types of experience, the blend makes a figure emerge that did not exist before.

When we compare the input spaces, we see basically the same type of mechanism at work behind different Forces of Nature. This allows us to deliberately produce analogical mappings that support both informal and formal reasoning about a phenomenon. We can use formal structures from theories of fluids in order to create formal models and theories of thermal processes. We can do the same with fluids and electricity, and we can finally use electrical formalisms in order to create thermal or fluid models (such as for heat transfer networks or the blood flow system of mammals, respectively; see [5]).

### 4.4. Carnot’s Waterfall Analogy as a Spontaneous Mental Activity

Calling Carnot’s analogy—which, as we have seen is intimately connected to the Caloric Theory—a spontaneous mental or cognitive activity raises two questions: How is it an activity? How can it be spontaneous? A table listing elements of an analogical mapping such as Table 1 is neither an activity nor was it drawn up spontaneously.

Calling it an activity rather than a structure still makes sense. Its meaning arises when engineers imagine heat flowing and residing in materials, making things hot, flowing spontaneously, i.e., falling from a hot to a cold place, possibly causing something to happen in the course, or, apparently, not doing much at all if we bring a hot and a cold body in direct contact or mixing a hot and a cold fluid; or when we imagine heat (caloric) being pumped. Its power lies in the imaginative activity it gives rise to which is central to creative engineering design, to comparing processes from vastly different fields, or to learning, for that matter.

The activity is spontaneous to a degree. It certainly is embedded in cultural practice. Parts of it are, however, decidedly spontaneous. The forming of schemas is not under the control of our will, nor is the act of metaphorical projection. It is true that we learn metaphorical linguistic expressions, and we control the willful construction of novel expressions; however, having access to and using conceptual metaphors in not a matter of wanting to do it or not [16,20]. Interestingly, gesture paralleling linguistic expression is quite spontaneous as well. And creating a blend of two experiential spaces through compression (Figure 4b) sometimes happens spontaneously, even though we can put it under our control as well—as Carnot demonstrated to us in his memoir, as W. and J. Thomson did in the 1840s (see Section 3.5), and as we shall do in Section 5.

In all this, the imagery of waterfalls is archetypal. It is not surprising that it should have arisen in science at some point of its history, and it is not surprising that it happened at the hands of engineers. It is rather surprising though, and worth considering, why the Waterfall Analogy faded into the background, even fell into oblivion. We shall not go into this here; we have discussed reasons for how the Caloric Theory came into disrepute [4], which is certainly one of the reasons, if not the most important one.

## 5. The Waterfall Analogy and Modern Macroscopic Physics

Now, we turn to generalizations and applications of Carnot’s Waterfall Analogy: we transfer the imagery, i.e., the schemas and metaphors, contained in it to the other six macroscopic processes treated in theories of uniform dynamical and continuous systems (see Table 2, and [5,9]). Then, we make use of what Carnot’s notion of power suggests about energy and construct an extended energy principle. Finally, we show how to create process diagrams for a handful of processes and systems, where we apply the principles of Forces of Nature constructed upon Carnot’s imagery.

### 5.1. Forces of Nature and the Waterfall Analogy in Macroscopic Physics

In theories of macroscopic processes and systems, we commonly treat what we have called Basic Forces of Nature: Fluids, Electricity (and Magnetism), Heat, Substances, Gravitation, Linear Motion, and Rotation (Table 2).

Every one of the Forces is characterized by an intensity (i.e., potential) and the related tension, an extensive quantity and its flows (and possible production and destruction rates), and power related to its fall or lifting (Figure 5). Since we are going to treat the following theme in a semi-formal manner, we shall switch to the technical term for caloric: we shall be using the term *entropy*; however, the reader may easily replace entropy by caloric (assuming the Extended Caloric Theory; [4,5]).

Importantly, there are three extensive quantities that are not conserved: fluid volume, amount of substance, and entropy. Amount of substance (and fluid volume) can be created and destroyed, and entropy can be created (Figure 6). This means we add the imagery of birth and death to the notions that allow us to generalize Carnot’s work on the theory of heat engines.

Doing so means we relate destruction of amount of substance (and fluid volume) to its fall from a value of potential equal to μ down to absolute zero: μ=0. The production of amount of substance and entropy (and fluid volume) is likened to lifting (pumping) these quantities from absolute zero (μ=0 and T=0) to appropriate levels of μ and T, respectively. Power is then expressed by the equations shown in Figure 6.

Note that the relations used for the power of a process apply to conductive flows of the extensive quantities, i.e., when the driving force of a process is the tension relating to the flow (see Table 2). Convective and radiative transfers need to be treated separately (see [5], chapters 7 and 8). (However, it is possible to treat convection and electromagnetic radiation as examples of chemical processes by extending the notion of chemical potential. We shall not go into this issue any further in this paper.)

Furthermore, assuming the expression for power of a process to be proportional to its tension may be understood as defining the notion of potential and potential difference. We can take the expression for gravitational power, i.e., $P_{grav}=∆\varphi_{G}I_{m}$, as introducing a general notion of potential $\varphi_{G}$, which, as we know, is proportional to height above ground only for a limited range of heights. In electricity, electric tension (voltage) is defined as depending upon the power of an electric current (flowing through this tension) and the magnitude of the current. Finally, the same holds for thermal and chemical potentials.

### 5.2. Constructing a Generalized Energy Principle: Extending the Waterfall Image

Let us now construct the most general energy principle that applies to macroscopic uniform dynamical systems and processes—visual metaphorical elements representing this generalized principle are shown in Figure 7. So far, we accept that the quantities we call power—those related to the fall or the pumping of fluidlike quantities such as entropy, charge, and angular momentum—suggest a first aspect of the concept of energy: power denotes the *rate at which energy is made available or used* in the coupling of Forces of Nature; see Equation (2).

Consider a heat engine which operates by entropy (caloric) falling from a high to a low temperature (Figure 7, left). If we do accept that this leads to energy being made available, we may ask where the energy comes from. Note that posing the question in this form means that we have already made a conceptual move: a person asking “where the energy comes from” and accepting that “energy can be made available”, assumes that energy is a metaphorical “thing” or “substance” [84,85,86,87] that carries the characteristics of an extensive quantity that can be “counted”.

Having clarified this, we suggest that the energy made available is not generated but delivered to the engine from the furnace. This brings up a second aspect of energy: it can be transported from element to element in a chain of devices and processes (Figure 7): there are energy currents. The transfer is affected by the entropy flowing from the furnace to the engine: entropy is an *energy carrier* [5,37]. Obviously, it carries some of the energy supplied out of the engine and into the environment. If the heat engine is running ideally (and only then!), the power of the process of a fall of entropy equals the difference of incoming and outgoing energy current (in general, it is calculated by Equation (3)):(6)$$P_{th}=I_{E1}-I_{E2}$$

What we observe about entropy, its flow from element to element as it carries energy, applies to all *conductive* flows of fluidlike quantities (Table 2). The linearity of the relation between power and potential differences can be shown to lead to
(7)$$I_{E}=\varphi_{X}I_{X}$$

Here, *X* denotes any of the processes listed in Table 2: $\varphi_{X}$ is the potential at which the carrier current $I_{X}$ occurs. Applying this to an Ideal Carnot Heat Engine (Section 3.3) having a (second law) efficiency equal to 1 allows us to immediately write down the first law efficiency of such a device (this is the traditional idea of the Carnot efficiency).

Note two things: First, for Electricity, Gravity, Rotation, and Linear Motion, the potentials are not absolute, which means that values of (conductive) energy currents calculated according to Equation (7) are relative to our choice of zero level or “ground”. This means that only the power of such processes carries independent meaning. Second, when the potential at which a conductive flow of fluidlike quantity occurs equals zero, there is no energy carried along (this is why we do not show energy currents on the return paths of angular momentum and charge in Figure 7).

Furthermore, the furnace and cooler may be looked upon as entropy and energy storage elements: this suggests that energy can be stored in physical objects; this is the third aspect of the generalized energy principle. The fourth is the well-known assumption of conservation of energy. In sum, this allows us to formulate a dynamical form of the law of balance of energy that does not include a production term:(8)$$\dot{E}=\sum I_{E,i}$$

Note that what we have described here leads to a corollary: the power terms associated with the coupling of processes in an element of uniform models (exemplified by process diagrams as in Figure 3 and Figure 7) balance (in the element symbolizing the dissipative heat engine in Figure 7, $P_{th}$, $P_{rot}$, and $P_{diss}$ add up to zero).

Equation (8) is well known; it summarizes the latter three aspects of energy: transfer, storage, and conservation. The first of the characteristics, the concept of power as we have taken it from Carnot’s Waterfall Analogy, is not known to the same degree; in fact, for most theories and models of classical physics it is simply unknown. In mechanics, power denotes energy transfer and not the quantity expressed in our Equations (1) and (3). We have to turn to continuum physics to find the concept of stress power [8,88] to find the (local) analog to Carnot’s and our concept of power of the fall of an extensive quantity.

Historically, formal steps have been taken at several points and have been instrumental for the development of the concept of energy in physical science. Famous among these steps is Clausius’ demonstration from his generic principles and constitutive relations for simple fluids that an energy function $E\left(V,T\right)$, i.e., internal energy, can be proved to exist ([61]; see [3], chapter 7; [5], chapters 5.3 and 10). Readers will have noticed that we did not follow this formal path but constructed the generalized energy principle by qualitative argument, i.e., by putting imagery and ideas before formal derivations. We stress that this imagery is part of natural forms of thought and that it may very well start from Carnot’s Waterfall Analogy.

### 5.3. Examples of Models

Here, we are going to construct models in the form of process diagrams where we apply Carnot’s imagery and its extensions. We shall treat (thermoelectric) heat pumps, a hydraulic power plant used to power lights, breaking of a flywheel, and the photovoltaic production and subsequent storage of hydrogen. As the last example, we return to heat engines and show how the model of an Ideal Carnot Heat Engine appears as part of the model of endoreversible engines.

In this and the following examples, we shall make use of the technique of process diagrams, as in Figure 3 and Figure 7. Process diagrams apply visual metaphors for the three fundamental aspects of Forces of Nature: intensities (potentials) and their differences; flows and production rates (and possibly storage) of extensive quantities; and quantities relating to the energy principle, i.e., power and flow (and possibly storage). In Figure 3, we have made use of power only, which means that we are true to Carnot’s imagery. In Figure 7, we have added the elements that make up the extended energy principle.

#### 5.3.1. Generic and Thermoelectric Heat Pumps

A generic heat pump, i.e., a dissipative device, can be understood by inverting the process diagram already shown in Figure 3b—naturally, the production of entropy cannot be reversed. In the diagram of Figure 8a, we have performed this inversion and added the flows of energy into and out of the device.

We might assume the heat pump to be powered by the fall of either angular momentum or electrical charge (depending upon what we take to be the driving Force). The energy supplied by the driving process is made available, after which it is used partly for pumping entropy, and partly for producing more entropy (which needs to leave the heat pump together with the entropy that is lifted from the cold environment). Since energy enters the device with entropy from the environment, the energy flow out of the pump is greater than the energy flow associated with the causing process.

In Figure 8b, we show a process diagram for a thermoelectric heat pump operated in a steady state (see [5,58,89]). There are flows of charge and entropy, and production rates of entropy. Note that dissipation results from the flow of charge and the conductive “back-flow” of entropy through the device from the warm to the cold space (note the index *c* in the entropy currents shown in the diagram). Pumping of entropy is the consequence of the flow of entropy coupled to the flow of charge (index *TE*), for which the power of part of the electrical process is responsible ($P_{el(TE)}$).

#### 5.3.2. Hydraulic Power Plant

Here, we demonstrate how to create a process diagram showing the chaining of devices and processes (Figure 9) using a number of uniform elements. Imagine an aggregate model for powering light in a community served by a hydroelectric power plant; the power plant makes use of water from an artificial lake in the mountains. The process diagram is again one of purely steady-state operation.

The story of the Forces coupling to each other starts with water falling from the mountain—we have a gravitational process making energy available. The energy is used partly for powering the Force of Rotation of a generator. The power of the caused rotational process equals the energy flow from the turbine to the generator, and the energy transferred is then made available in the fall of angular momentum. Again, part of the energy made available in this second fall is used for pumping electrical charge which, finally, in its fall, powers the production of Light (notice how the final process—production of Light—is abbreviated and represented as a convective flow of Light which transports entropy as well).

#### 5.3.3. Breaking of a Flywheel

Here is an example of a dynamical process: the breaking of a flywheel (Figure 10). The motion of the flywheel is decelerated with the help of a generator which is part of a closed circuit containing an electric resistor [90].

In the model, the flywheel itself does not serve as a coupler for (two or more) Forces of Nature, unlike the generator and the external resistor—it is simply a storage element for angular momentum and energy. Angular momentum will flow out of the flywheel at its instantaneous high angular speed, from where it will fall to a level equal to zero (“ground”) in the generator. In the fall, energy is made available, which is used for pumping electrical charge. The electrical charge will then fall in two steps back down to its original level and so power two processes: production of entropy in the internal electrical resistive element and the external resistor.

A slightly more sophisticated model includes the parallel operation of falling angular momentum in a mechanical dissipative layer (friction layer). A part of the angular momentum of the flywheel will leave not through the generator but through the friction layer between rotating elements, casings, and the ground. Again, energy is made available in the fall of angular momentum. Since the friction layer is purely dissipative (as are the electric resistors), all the energy made available in this fall is dissipated: entropy is produced.

#### 5.3.4. Sun Electrolyzer, and Hydrogen Storage

Let us see how to apply Carnot’s imagery of waterfalls and the pumping of water to processes involving chemical reactions (for detailed treatment of another example of chemical reactions, see [70]). What we learn here is that, if we take the difference of chemical potentials, i.e., chemical tension, between reactants and products as the drive of a chemical reaction [91,92], we encounter another case of the Waterfall Analogy.

In the chain of processes depicted in Figure 11, both solar cells and the electrolyzer are couplers where chemical reactions occur: we are treating the absorption of light and the production of light in the solar cells as a chemical process [93]; the electrolysis of water occurring in the electrolyzer serves as a more typical example of a chemical reaction, as would be its inverse occurring in a hydrogen fuel cell [94].

Consider the absorption of Sunlight in the solar cells; at the same time, the cells produce (infrared) light. We may consider the process the destruction of light-as-a-substance which makes energy available (Figure 6a); production of light-as-a-substance, on the other hand, uses (a part of this) energy (Figure 6b)—the difference relates to the power of the total chemical process ($P_{ch}$), and the associated amount of energy is made available for pumping electrical charge ($P_{el}$) and producing entropy ($P_{diss}$). The absorption and emission of light constitute a spontaneous chemical reaction that makes energy available.

In the electrolyzer, we confront a non-spontaneous (driven) chemical process, namely the production of Hydrogen and Oxygen from Water (note, we treat every one of these substances as an independent Force of Nature, having its characteristics of potential, amount of substance, end energy).

#### 5.3.5. Models of Endoreversible Systems and Processes

Let us revisit Carnot’s original model and see how it can be used and extended to include dissipation (Figure 12). As was clear to Carnot, the major contribution to irreversibility, his “true loss of power”, had to stem from “useless re-establishment of equilibrium in the caloric”, i.e., the (conductive) flow of caloric (entropy) resulting from direct contact of a warm and a cold body. In a heat engine, two such contacts are necessary for a finite flow of caloric: one between the furnace and the working fluid, and one between the latter and the cooler.

An Ideal Carnot Heat Engine, on the other hand, would receive and reject entropy at fixed upper and lower temperatures of the working fluid and lower the entropy reversibly between these levels. The process diagram of a model for the complete system is shown in Figure 12. Simply put, entropy flows from the furnace through a first heat exchanger (where additional entropy is produced) into and through the ICHE (where the entropy is let fall reversibly, thus “creating motive power”), out of the ICHE through a second heat exchange (where additional entropy is produced), and from there into the cooler.

Models of this type have been called endoreversible—models of internally reversible engines [5,33,39,40,95]. There is an interesting feature of such models: they can be optimized by assuming that, in steady-state operation, the overall production rate of entropy will be minimal. Such models have been produced for generic heat engines or thermal powerplants (leading to the Curzon–Ahlborn–Novikov efficiency; [39,96,97]; [5], chapter 9.2), for the generation of Wind at the surface of a planet ([95,98]; [5], chapter 9.6), concentrating solar thermal engines ([95]; [5], chapter 9.3), and photovoltaic cells [95], among others.

## 6. Summary and Conclusions

In the previous sections, we demonstrated that—apart from assumptions concerning specific constitutive quantities and relations, such as the ideal gas law—Carnot’s thinking was founded on two premises which form a tight unity: the Waterfall Analogy and the Caloric Theory of Heat (CTH). We had investigated the background of the CTH in a previous paper [4] where we showed in what sense the extended notion of caloric is analogous to the concept of entropy in modern macroscopic thermodynamics, both in theories of uniform dynamical systems and in continuum thermodynamics [5].

Here, we accept Carnot’s fundamental assumptions and explore where they lead us. In Section 2, we explicated the cognitive and formal structures of the waterfall imagery as part of the concept of Force of Nature and made the transfer to Carnot’s work on heat engines (Section 3). Furthermore, we briefly investigated the origin of the Waterfall Analogy and asked how it fared in the 200 years since the publication of *La puissance du feu* [1]. To the usual assumptions concerning sources (i.e., Carnot’s father, practice in the engineering of hydraulic machinery, etc.), we added the suggestion that the Waterfall Analogy is a natural form of human understanding; we presented the theoretical underpinnings for our claim that waterfalls emerge as an archetype of human experiencing that is applied spontaneously in analogical structure mapping (Section 4). Its appearance in physical science should not surprise us; rather, we should wonder how little use has been made of Carnot’s imagery in the years since it was described.

In this paper, we have made the notion of *power of a fall* (of amount of water or amount of heat/caloric) centrally important: we have made it explicit by deriving from it the concept of *energy made available* in a spontaneous process. Starting from this, we can construct a generalized energy principle for macroscopic physical systems and processes and apply the Waterfall Analogy (and its reverse: the pumping of fluidlike quantities) to all the phenomena dealt with in theories of uniform dynamical systems and continuum physics (Section 5). Importantly, the imagery of fall and lifting (pumping) is extended to processes where extensive quantities are created and destroyed (amount of substance in chemical processes) or created (caloric/entropy in dissipative processes). A number of concrete examples show how Carnot’s imagery can serve as a powerful tool for analogical transfer. Both the generalized energy principle and the examples demonstrate the power of Carnot’s waterfall imagery for science and engineering.

Two hundred years after Carnot made us aware of it, his Waterfall Analogy is a powerful example of theory construction with roots deep in how we experience phenomena as caused by natural agents.

## Figures and Tables

**Figure 1 entropy-26-01066-f001:**
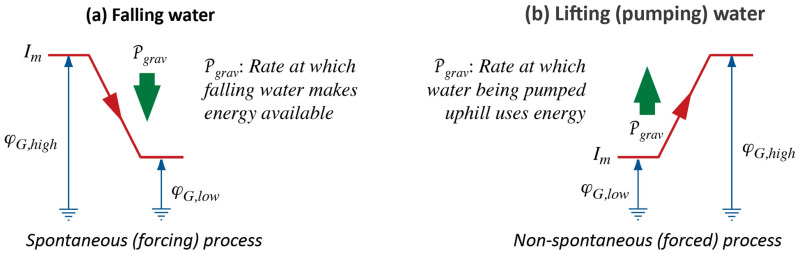
Falling water and pumping (lifting or raising) water. The former process is spontaneous: it forces (causes) some follow-up processes. The latter is non-spontaneous: it needs to be forced or driven. Here, $\varphi_{G}$ is the gravitational potential, $I_{m}$ denotes the current of mass of water, and $P_{grav}$ is the power of the gravitational process, for both spontaneous and non-spontaneous cases.

**Figure 2 entropy-26-01066-f002:**
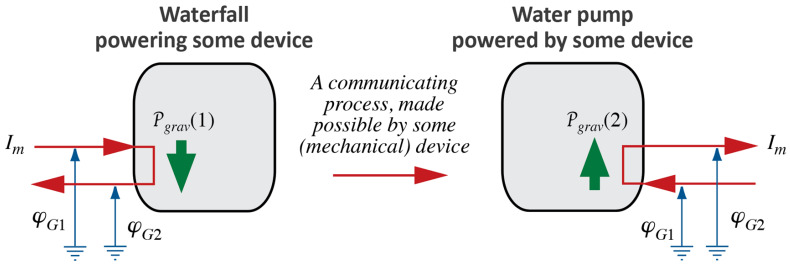
Falling water and pumping (lifting or raising) water. The former process is spontaneous: it forces (causes) some follow-up processes. The latter is non-spontaneous: it needs to be forced or driven. Here, $\varphi_{G}$ is the gravitational potential, $I_{m}$ denotes the current of mass of water, and $P_{grav}$ is the power of the gravitational process, for both spontaneous and non-spontaneous cases.

**Figure 3 entropy-26-01066-f003:**
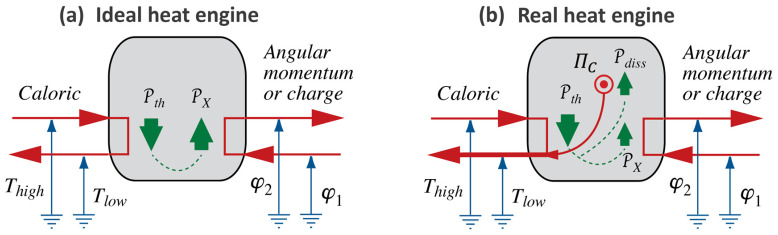
Process diagrams of an ideal (**a**) and a real (**b**) heat engine. In either case, a current of caloric falls from $T_{high}$ to $T_{low}$, and the thermal power $P_{th}$ is the same. However, what happens then is different: in (**a**), all the energy made available by falling caloric is used for pumping a fluidlike quantity such as angular momentum or electric charge ($P_{X}=P_{th}$). In (**b**), part of the energy made available is not used for the desired process; rather, it is used for producing more caloric that flows out of the engine at $T_{low}$. The circle with dot in (**b**) denotes a production rate of caloric $\Pi_{C}$, and the dashed lines symbolize the “handing over” of energy.

**Figure 4 entropy-26-01066-f004:**
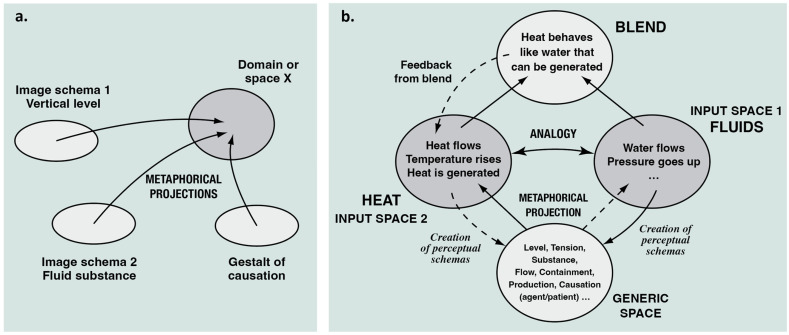
(**a**) Diagram suggesting how structure inherent in image schemas (which serve as source domains) is projected metaphorically onto a target domain X. (**b**) Diagram showing conceptual integration at work in fluids and heat. Fluids and, to a lesser extent, heat create elements of a generic space that is used in metaphoric renderings of theories of fluids and heat. Since both types of experience use mostly the same webs of metaphors, input spaces 1 and 2 appear similar to the human mind (adapted from [28]).

**Figure 5 entropy-26-01066-f005:**
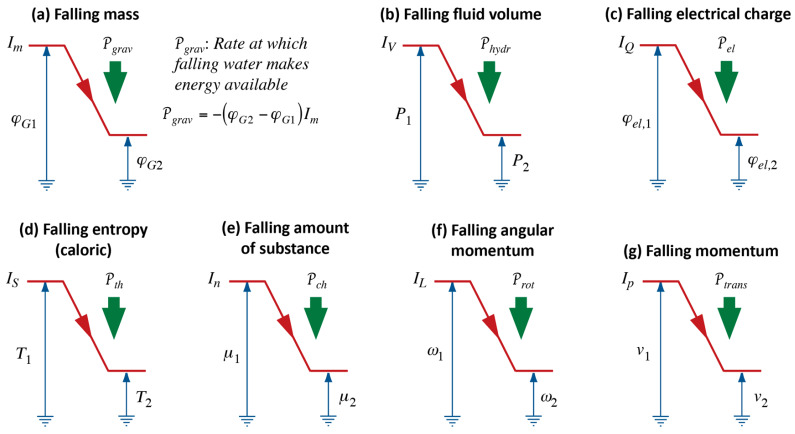
Visual metaphors of falling fluidlike quantities (the seven extensive quantities listed in Table 2), and associated potentials. Each of the diagrams can be reversed: the flow of the fluidlike quantities can be forced to go from low to high potential. Adapted from [5].

**Figure 6 entropy-26-01066-f006:**
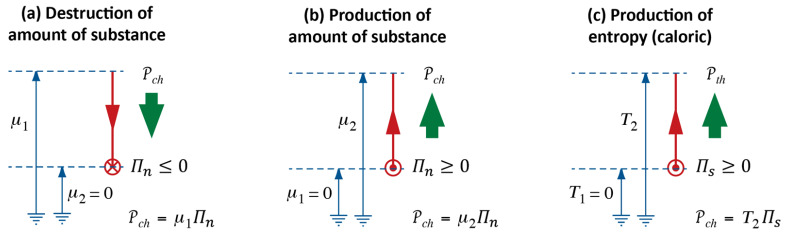
(**a**) Fall and destruction of amount of substance of educts in a chemical reaction. (**b**) Production and lifting (pumping) of products in a chemical reaction (in (**a**,**b**), absorption and emission of light are included). (**c**) Production and lifting of entropy (caloric) in a dissipative process. We exclude fluid volume here; if desired, hydraulic processes can be subsumed under chemical phenomena.

**Figure 7 entropy-26-01066-f007:**
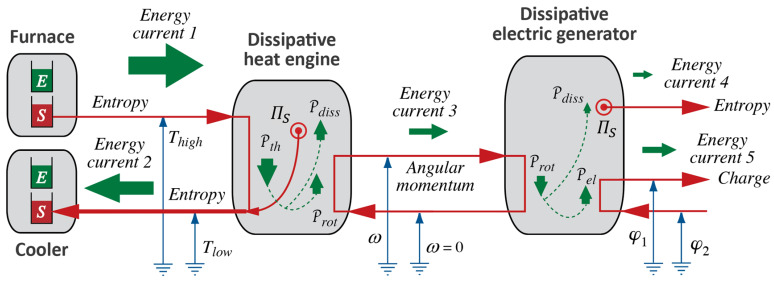
Process diagram of a real heat engine (pumping angular momentum) driving a real generator (pumping electrical charge). Entropy for the heat engine comes from the furnace and is rejected to the cooler (together with the entropy generated in the heat engine). Transfers of entropy, angular momentum, and charge are accompanied by energy currents. The storage of entropy in the furnace and the cooler is coupled to energy stored in these bodies.

**Figure 8 entropy-26-01066-f008:**
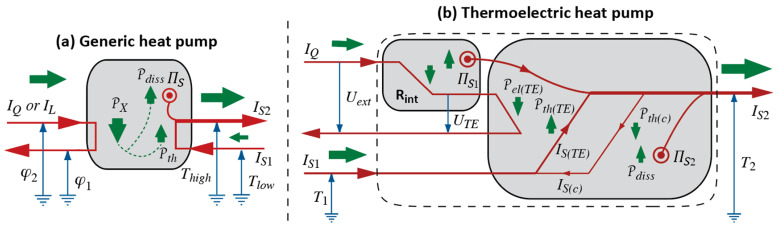
Steady-state process diagrams of (**a**) a generic and (**b**) a thermoelectric heat pump (adapted from [5], chapter 4). The generic heat pump diagram is the inverse of the (dissipative) heat engine diagram shown in Figure 3b. (**b**) Details of internal processes that allow us to pinpoint sources of dissipation; here, they are electrical and thermal due to the conductive (index (**c**)) flow of entropy from the hot reservoir back through the heat engine. Lifting of entropy ($I_{S(TE)}$) is the consequence of the thermoelectric process.

**Figure 9 entropy-26-01066-f009:**
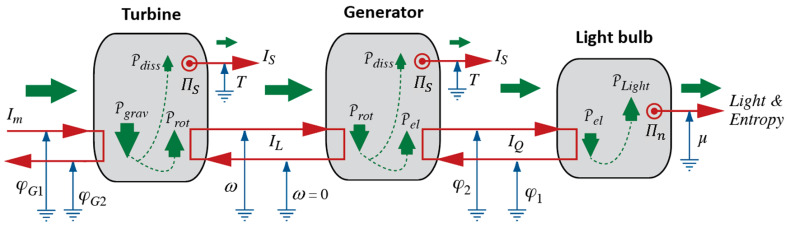
Process diagram for a chain of elements and processes: turbine, generator and light bulb allow for the coupling of Gravity to Electricity to Light. Note that the details of coupling of Electricity and Magnetism in the generator are hidden by creating an aggregate uniform object called “generator”. A similar process of aggregation is used for the coupling of Electricity and Light in the light bulb.

**Figure 10 entropy-26-01066-f010:**
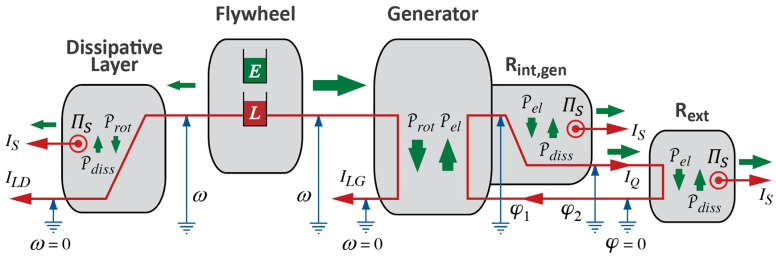
A process diagram for a dynamical process—the breaking motion of a flywheel—must include symbols for storage, here of angular momentum and of energy. Both quantities flow out of the flywheel. In the end, the total energy that was stored will be dissipated in the production of entropy.

**Figure 11 entropy-26-01066-f011:**
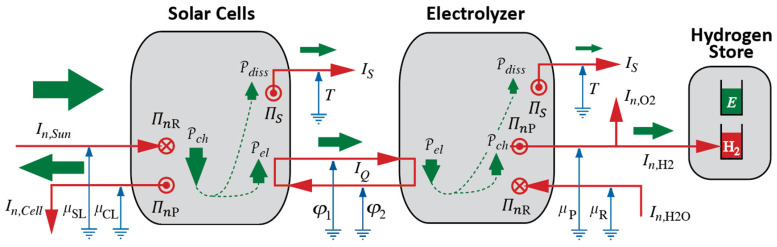
A chain composed of solar cells, electrolyzer, and hydrogen storage where Light, Electricity, Water, Hydrogen, and Oxygen are active as Forces of Nature.

**Figure 12 entropy-26-01066-f012:**
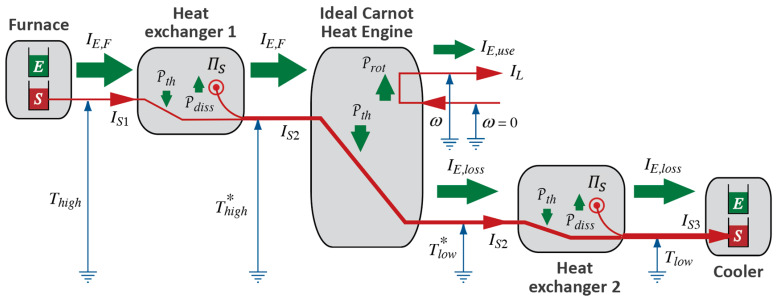
Process diagram of an endoreversible heat engine: the actual engine sandwiched between heat exchangers is assumed to be an Ideal Carnot Heat Engine.

**Table 1 entropy-26-01066-t001:** Waterfall Analogy: Some important mappings between Water and Heat.

Water, Waterfalls, and Water Wheels	Heat, Falling Caloric, and Heat Engines
Vertical level, level difference	Temperature, temperature difference
Gravitational tension	Thermal tension
Amount of water (mass of water)	Amount of heat, i.e., caloric
Flow/fall of water	Flow/fall of caloric
Amount of water contained in a system	Heat (caloric) contained in materials (assumption of a heat function)
Conservation of amount of water	Conservation of quantity of caloric
Power of falling water (falling water produces motive power)	Power of falling caloric (falling caloric produces motive power)
Power is proportional to both gravitational tension and flow of water	Power is proportional to both thermal tension and flow of caloric
Loss of power of falling water (water falling without “producing motive power”)	Loss of power of falling caloric (caloric falling through a temperature difference without “producing motive power”)
Water falling and water wheel	Caloric falling and heat engine
Water can be lifted/pumped (brought back to a high level)	Heat (caloric) can be pumped (brought back to high temperature)
Power is independent of medium falling	Power is independent of working fluid
Water pump	Heat pump
Impossibility of hydraulic perpetual motion machine	Impossibility of thermal perpetual motion machine

**Table 2 entropy-26-01066-t002:** Fields of macroscopic physics as Basic Forces of Nature: intensity, extension, and symbols.

Force of Nature	Intensity		Extension	
Fluids	Pressure	P	Volume	V
Electricity	Electric potential	φ	Charge	Q
Heat	Temperature	T	Entropy (Caloric)	S
Substances	Chemical potential	μ	Amount of substance	n
Gravitation	Gravitational potential	$\varphi_{G}$	Gravitational mass	m
Linear Motion	Velocity	v	Momentum	p
Rotation	Angular velocity	ω	Angular momentum	L

## Data Availability

The original contributions presented in this study are included in the article. Further inquiries can be directed to the corresponding authors.

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
