# Peer review of "Carnot and the Archetype of Waterfalls"

_entropy, 2024, doi:10.3390/e26121066_

Round 1
Reviewer 1 Report
Comments and Suggestions for Authors
Attached pdf file.

Reviewer 2 Report
Comments and Suggestions for Authors
We all agree that "The waterfall analogy" became for Sadi Carnot a starting point to reflect on the question: how do heat engines work? This analogy was suggested by Lazar Carnot in 1815. It was in Warsaw, where General Lazar Carnot was in exile and where his son Sadi visited him. Legend has it that they visited the Warsaw waterworks, where Watt's steam engine worked, and then the father asked his son: how does this engine work?
The article by Prof. Fuchs and colleagues perfectly meets the needs of the continuous construction of the foundations of thermodynamics. In Poland, many physicists accuse thermodynamics of being a science without foundations, that instead of foundations and powerful bases it is based only on myths, analogies and archetypes. I was really happy to receive an article that discusses not only the tools but also the ways of thinking that lead us to discover the main laws of nature.
Prof. Fuchs' article is complete, written in great language, inspiring - and the illustrations are convincing. It is a good starting point for considering the question: is heat the fifth force of nature?
I recommend the article for printing in ENTROPY
